

# The role of mechanical stratigraphy on the refraction of strike-slip faults

Mirko Carlini[1*], Giulio Viola[1], Jussi Mattila[2], Luca Castellucci[1]

1 BiGeA – Department of Biological, Geological and Environmental Sciences, University of Bologna, Italy

GTK – Geologian Tutkimuskeskus, Geological Survey of Finland, Espoo, Finland

*Corresponding author: mirko.carlini2@unibo.it

**Abstract.** Fault and fracture planes (FFP) that cut through multilayer sequences can be significantly refracted at layer-layer interfaces due to the different mechanical properties of the contiguous layers, such as shear strength, friction coefficient and grain size. Detailed studies of different but coexisting and broadly coeval failure modes (tensile, hybrid and shear) within multilayers deformed in extensional settings have led to infer relatively low confinement and differential

stress as the boundary stress conditions at which FFP refraction occurs. Although indeed widely recognized and studied in extensional settings, the details of FFP nucleation, propagation and refraction through multilayers remain not completely understood, partly because of the common lack of geological structures documenting the incipient and intermediate stages of deformation. Here we present the results of a study on strongly refracted strike-slip FFP within the mechanically layered turbidites of the Marnoso Arenacea Formation (MAF) of the Italian Northern Apennines. The MAF

is characterized by the alternation of sandstone (strong) and carbonate mudstone (weak) layers. The studied refracted FFP formed at the front of the regional-scale NE-verging Palazzuolo anticline and post-date almost any other observed structure except for a set of late extensional faults. The studied faults display coexistence of shear and hybrid (tensile-shear) failure modes and we suggest that they initially nucleated as shear fractures (mode III) within the weak layers and, only at a later stage, propagated as dilatant fractures (mode I-II) within the strong layers. The tensile fractures within the

strong layers invariably contain blocky calcite infills, which are, on the other hand, almost completely absent along the shear fracture planes deforming the weak layers. Paleostress analysis was performed to constrain the NNE-SSW compressional stress field that produced the refracted FFP and to exclude the possibility that the present attitude of these structures may result from the rotation through time of faults with an initial orientation. Slip tendency analysis was also performed to infer the relative slip and dilation potentials of the observed structures. Mesoscopic analysis of preserved

structures from the incipient and intermediate stages of development and evolution of the refracted FFP allowed us to build an evolutionary scheme wherein: a) Nucleation of refracted FFP occurs within weak layers; b) Refraction is primarily controlled by grain size and clay mineral content and variations thereof at layer-layer interfaces but also within individual layers; c) Propagation within strong layers occurs primarily by fluid-assisted development ahead of the FFP tip of a "process zone" defined by a network of hybrid and tensile fractures; d) The process zone causes the progressive

weakening and fragmentation of the affected rock volume to eventually allow the FFP to propagate through the strong layers; e) Enhanced suitable conditions for the development of tensile and hybrid fractures can be also achieved thanks to the important role played by pressured fluids.





## 1 Introduction

Refraction of fault and fracture planes (FFP) is defined as a significant change of their trajectory due to their crossing of rocks characterized by layered mechanical properties. This may reflect, for instance, variations in lithology (composition and/or grain size) and degree of compaction.

The refraction of FFP has been well documented and studied particularly in extensional settings, where it has been convincingly shown that where normal faults cut through multilayer systems they are refracted due to the differential mechanical properties of the juxtaposed faulted layers (e.g., Ferrill et al., 2017 and references therein; Giorgetti et al., 2016; Agosta et al., 2015; Schoepfer et al., 2006; Sibson, 2000). The heterogeneity of the mechanical properties within such systems is essentially the result of alternating strong and weak layers (in terms of shear strength and friction

coefficient) and causes the spatially-partitioned coexistence of tensile (mode I) and hybrid (mixed mode I-II) failure modes within the strong layers and shear failure (mode III) within weak layers.

The stress conditions necessary for the development of refracted FFP generally require relatively low confining pressure (e.g. Schoepfer et al., 2006) and low differential stress (Ramsey and Chester, 2004; Ferrill et al., 2012). In particular, mode I failure is generally achieved with a differential stress < 4T (T: tensile strength), while hybrid failure takes place

for differential stress magnitudes > 4T but <5.66T (Secor, 1965; Sibson, 2003). Failure mechanisms leading to refraction require the development and coexistence of a complex and locally transient state of stress that causes each involved lithology to behave mechanically differently and to follow partially independent and not fully synchronous strain paths. When subject to a given differential stress, strong layers thus tend to deform elastically until failure, while weak layers accumulate strain inelastically (e.g. Giorgetti et al., 2016). The differential mechanical behaviour of a multilayer is also

reflected in the differential involvement of fluids such that, while fluids, when present, may deeply impact upon the mechanical strength of strong layers, they barely affect the inelastic behaviour of weak layers, which deform under essentially undrained conditions (e.g. Rudnicki, 1984).

One controversial and debated aspect concerning FFP refraction is whether the rupture nucleates within the strong or the weak layers and the details of subsequent propagation across the multilayer. Nucleation within strong layers requires

initial strain localization by formation of extensional fractures at high angle to the layer that subsequently physically interconnect by thoroughgoing shear fractures and end up affecting also the weak layers (e.g., Peacock and Sanderson, 1995; Sibson, 1998; Ferrill and Morris, 2003; Schoepfer et al., 2006). On the other hand, nucleation within weak layers requires initial nucleation and localization by growth of shear fractures and the later opening of connected dilational fractures within the strong beds (e.g. Peacock and Zhang, 1994; Roche et al., 2013; Agosta et al., 2015; Giorgetti et al.,

2016). As to the post-nucleation propagation of refracted FFP, the most commonly invoked model contemplates their evolution in extensional settings. Refracted FFP are thus interpreted as extensional faults whose development occurs under overall shear failure forming pull-aparts or extensional/releasing bends (e.g. Peacock and Zhang, 1994 and Peacock and Sanderson, 1995; Hill-type fractures by Hill, 1977; Sibson, 1996). Alternatively, however, some models suggest the presence of already refracted individual shear segments (staircase geometry) at the onset of refracted FFP development,

that cause the opening of later, intervening dilational jogs (Ferrill and Morris, 2003; Sibson and Scott, 1998; Giorgetti et al., 2016).

Discriminating between these conceptually different geometric and mechanical models is commonly hampered by the paucity of well-preserved field evidence of structures documenting the incipient and/or intermediate evolution stages of refracted FFP. A unifying mechanical model describing the details of the formation and propagation of hybrid faults and

fractures and the transition from tensile to hybrid and to shear fractures is, thus, still incomplete (e.g. Ramsey and Chester, 2004). Moreover, indeed because of the lack of solid documentation of the early and intermediate evolutionary stages, it cannot be excluded that the same final geometry may even result from different processes.



To further contribute to the study of these mechanical aspects, we have analysed well-exposed refracted strike-slip FFP in alternating sandstone/mudstone beds of flysch deposits of the Northern Apennines (Italy). The presence of structures testifying to the incipient and intermediate stages of refraction and clear evidence of the mechanical role played by the different lithologies and fluid involvement upon the differential mechanical behaviour of strong and weak layers, has

allowed us to document the details of the evolution of these structures through space and time and to suggest constraints upon their nucleation and propagation history.

From a mechanical viewpoint, the observed nucleating structures allowed us to better understand how the angle $\theta$ at which fractures form to the orientation of the principal compressive stress $\sigma_1$ controls the failure mode, and to study a natural example of transient strain evolution, from tensile to shear conditions.

**2 Geological Framework**

The study area is located in the Northern Apennines of Italy, on the northeastern-facing slope of the chain, where the local geology is dominated by the Marnoso Arenacea Formation (MAF; Fig.1, Benini et al., 2014). The MAF represents the infill of the Aquitanian-Messinian Apenninic foredeep basin (Ricci Lucchi, 1986; Roveri et al., 2002; Tinterri and Tagliaferri, 2015) and is characterized by regularly alternating sandstone and mudstone layers. The Alps-derived MAF

reaches a maximum thickness of ~5000 m and yields an average sandstone/mudstone thickness ratio of ~1:3 (Fig. 2). The studied portion of the MAF was affected by mid-to-late Miocene syn-depositional tectonic shortening, partially responsible for the closure of the foredeep basin and the development of the Palazzuolo anticline, one of the largest and best outcropping contractional structures of the Northern Apennines. The Palazzuolo anticline developed at a relatively shallow crustal level recording syndeformational T< ~100-110°C (Carlini et al., 2017) and is interpreted as a fault-

propagation fold, whose latest development relates to the activity of the nearby out-of-sequence Mt. Castellaccio thrust probably during the Messinian-Pliocene time interval (Carlini et al., 2017; Landuzzi, 2004). During the Late Miocene, syn-contractional low-angle extensional tectonics started to affect the Northern Apennines orogenic wedge at different structural levels (Molli et al., 2018; Jolivet et al., 1998) dramatically reshaping the belt compression-related architecture. In the study area, tectonic thinning was mainly accommodated within the Sestola Vidiciatico Unit (SVU), a ~500 m thick

shear zone separating the uppermost ocean-derived late Cretaceous-to-mid Eocene Ligurian Units from the underlying foredeep deposits of the MAF (Bettelli et al., 2012). Since the Pliocene, most of the chain has been affected by the still ongoing seismogenic high-angle extensional tectonics at shallow structural levels (<15 km), as testified by seismicity and GPS data (Eva et al., 2014; Bennett et al., 2012; Cenni et al., 2012). Only the deepest portion (>15 km) of the orogenic wedge and the outermost domains of the Northern Apennines (buried beneath the Po plain), instead, are still affected by

a NE-SW oriented and seismogenic thrusting regime. Still poorly understood strike-slip faulting, moreover, affects selected portions of the wedge and has been tentatively connected with the deep dynamics of the subducting slab (e.g. Piccinini et al., 2014 and references therein).

Our study concentrated on the well exposed northeastern front of the Palazzuolo anticline along the right bank of the Santerno river, on a c. 200 m long and 10 m wide outcrop (Fig. 2). In the study area, MAF layers are characterized by an

average thickness of c. 15 cm and a slightly overturned attitude (average bedding attitude 54/223-dip/dip direction), due to folding. The average composition of MAF layers is given by a feldspathic/lithic-rich detrital component (Quartz-57.8, Feldspar-24.6, Lithics-17.6) and a fine-grained lithic component containing equal contributions of metamorphic (Lm) and sedimentary/carbonate (Ls) fragments and a lower content in volcanic fragments (Lv) (Lm52.3, Ls42.5, Lv5.2; Benini et al., 2014; Fig. 3). Ls fragments are partially constituted by different amounts of phyllosilicates such as muscovite, illite,

kaolinite and chlorite. Experimentally-determined mechanical parameters are available for both lithologies, yielding similar values for the sandstone (shear strength C = 19 MPa, friction coefficient $\varphi$ = 46°) and the mudstone (C = 13 MPa; $\varphi$ = 40°; Lembo Fazio and Ribacchi, 1990).





**3 Used Methods**

**3.1 Paleostress analysis**

Numerous striated fault planes were studied and characterized at the outcrop by systematically collecting fault-slip data. A classic paleostress analysis was performed by the analytical approach offered by the Win-Tensor software (Delvaux

and Sperner, 2003), which inverts fault kinematic data to compute the reduced stress tensor that best accounts for a given fault population, fully defined by the orientation of $\sigma_1$, $\sigma_2$, $\sigma_3$ and R - the stress shape ratio, defined as

$$R = \frac{\sigma_2 - \sigma_3}{\sigma_1 - \sigma_3}$$

The best fit between the calculated stress tensor and the observed fault-slip data is obtained by a progressive rotation of the computed stress tensor to minimize the misfit angle $\alpha$ between the orientation of the observed fault striae and the maximum computed shear stress resolved on the fault plane. The optimisation of the misfit angle is obtained by minimising the normal stress and maximising the shear stress theoretically acting on the studied fault plane. In our study the maximum acceptable misfit angle was set to 30°, as suggested by Ramsay and Lisle (2000). Additionally, we

calculated the modified stress regime R' parameter (Delvaux et al., 1997) because it univocally identifies the stress regime resulting from the paleostress analysis with a number ranging from 0 to 3 and is directly derived from the stress ratio R as shown in Fig. 4.

**3.2 Slip tendency analysis**

Slip tendency (Morris et al. 1996) is defined as the ratio of resolved shear stress to resolved normal stress on a fracture

plane and slip tendency analysis can be employed in the assessment of the fracture orientations that are the most likely to develop by shear fracturing in a given stress field. A prerequisite for the analysis is, obviously, that the stress tensor is known. Dilation of fractures, on the other hand, is controlled by the normal stress, and a qualified prediction of the most likely orientations for fracture dilation can be generated by dilation tendency analysis. Dilation tendency (Dt) is defined by Ferrill et al. (1999) as:

$Dt = \frac{\sigma_1 - \sigma_n}{\sigma_1 - \sigma_3}$

The orientation of the principal stress axes and the corresponding stress ratio (R) were derived from paleostress analysis and have been used to calculate slip and dilation tendencies with a Matlab® script following the method proposed by Lisle and Srivastava (2004).

**4 Results**

**4.1 Field data**

Within the analysed area, the MAF deformed in a completely brittle manner and, according to our interpretation, contains faults that can be ascribed on a geometric and kinematic basis to five different fault families: (1) SE steeply-dipping refracted left-lateral strike-slip faults; (2) SW steeply-dipping right-lateral strike-slip faults; (3) SW steeply-to-shallowly dipping reverse faults; (4) SW-dipping bed-parallel faults and (5) N-to-NE and W-to-SW shallowly-to-steeply dipping

low- and high-angle normal faults (Fig. 2). None of the observed fault families contain a significant damage zone, not even within the weak mudstone layers (Fig. 5). Fault cores within the weak layers are usually very thin, less than 1 cm thick, and are occasionally coated with calcite. For the more mature faults, however, calcite forms thicker infills (~1-5




cm; Fig. 5D). In the case of the most significantly refracted faults, the thickness of the fault core changes dramatically as a function of the failure mode observed within the different mechanical layers (see section 4.3 for details; Fig. 5).

The relative chronology of the faulting events in relation to the development of the Palazzuolo anticline and the Mt. Castellaccio thrust remains loosely constrained, even though almost all the observed faults appear to postdate these regional structures. No clear crosscutting relationships among the different faults families were observed, with the exception of the faults that clearly reactivate the tensional fractures (that patently postdate the refracted faults) and the high-angle extensional faults that cut and postdate all other observed brittle structures.

**Mesoscopic analysis of refracted strike-slip faults**

The analysed strike-slip FFP are mostly left lateral, but a few dextral planes also occur. All these FFP exhibit a strongly refracted geometry in correspondence of the compositional/mechanical interface between weak (mudstone) and strong (sandstone) layers (Figs. 5A, 5B, 5C). Their observable displacement progressively decreases from the strong to the weak layers, where their tips are located and the FFP splay into a typical horsetail geometry (Fig. 5A). Within the mudstone layers deformation is more diffuse and is accommodated by a fracture cleavage mostly oriented at high angle to the bedding and to the localized refracted faults. These fractures are almost entirely barren of calcite infill (Fig. 5A).

Hybrid tensile-shear fractures are invariably found within the strong layers and are characterized by an aperture of between 4 and 20 cm and accommodate a lateral offset from 1 to 20 cm (Figs. 5A, 7). They contain blocky euhedral calcite without any indication of progressive and/or directional opening. Seldomly they bear slip indicators on the high-angle surface between calcite infill and host rock indicating a component of shear acting after fracture development and calcite infill. The thickness of the fault core of the refracted faults is usually less than 0.5 cm in mudstone layers, but it abruptly increases in the sandstone layers. The strong layers locally exhibit evenly spaced vertical tension gashes (i.e. dilatant calcite-filled fractures elongated along the vertical $\sigma_2$ direction) that are neither interconnected nor connected to other shear fractures/faults. The main shear FFP usually develop at about 30° to $\sigma_1$ (Fig. 5A) and they do not contain any interconnected "en échelon" segments.

The thickness of the sandstone and mudstone layers and their grain-size both appear to play a role in controlling the geometry of the refracted FFP. Concerning the thickness, the acute angle between the refracted FFP and bedding within mudstone layers varies between 60° and 30° (average 45.5°), as a function of the thickness of the layers. The trajectory of the FFP is observed not to be affected by the competence contrast between neighbouring layers when the thickness of the sandstone layer is <5 cm. Nonetheless, this first-order relationship appears to be locally overruled by the effect played by the grain-size of the deformed lithotype. In some cases, for example, FFP are refracted even at the interface between mudstone and 3 cm-thick coarse-grained sandstone layer, while, in other cases, fine-grained sandstone layers up to 10 cm thick are cut by throughgoing faults without any refraction. When the total amount of displacement is > ~50 cm, i.e. when the faults evolved toward a more "mature" stage (see discussion below), the thickness of the fault core increases while the refracted geometry tends to become less evident and, even if present in a first stage of the faulting activity, it is subsequently obliterated by the increase in displacement along the fault plane (Fig. 5D).

Several lines of evidence constrain the progressive spatial evolution of the refracted FFP through the multilayer. FFP exhibit well-developed shear planes within the mudstone layers that propagate also through the sandstone layers, but without completely cutting them through (Fig. 6). In some cases, the slip plane refracts at the mechanical interface between weak and strong layers, becoming almost at right angle to the bed, and is deformed by calcite-filled tensile fractures within the finest portion of the sandstone layers (Figs. 6A-6B). Still within strong layers, however, these structures exhibit a second-order refraction in domains where the grain size changes quite abruptly in the sandy beds, producing a branching geometry in the coarsest portion of the layers, characterized by the presence of both tensile and hybrid fractures (Figs. 6A-6B). These two types of fractures have been identified on the basis of θ (angle between $\sigma_1$ and





fracture), it being close to 0° for the Mode I openings (i.e. fractures parallel to the maximum compressive stress) and between 0 and 30° for the hybrid shear planes. In other cases, faults propagating through the sandstone layers produce a complex network of calcite-filled fractures with angular and not-sorted fragments of rock that are only partially dislodged from the host rock (Fig. 6C).

5 We derived geometrical scaling relationships to better characterize the tensile and shear components of the hybrid fractures within the strong layers. Fig. 7 plots the thickness of strong layers vs. the aperture of the dilational segments of the refracted FFP (tensile component) vs. offset (shear component). Although with a significant variance (see the low correlation coefficient), the data indicate a weak linear correlation for all data sets. The main reason for this dispersion is that the measured aperture and offset, as previously stated, strongly depend also on grain-size and clay mineral content, 10 which vary, at times significantly even within the same layer.

### 4.2 Paleostress analysis

Paleostress analysis was performed to constrain realistic stress fields capable of accounting for all the observed faults and to elucidate the possible genetic relationships between the computed paleostress tensor and the regional stress field that produced the Palazzuolo anticline within the overall North Apennines shortening framework. We did not include in this 15 analysis the high angle extensional faults because they are interpreted as the most recent tectonic features, basically unrelated to the other studied faults (see also section 2).

Results suggest that all the documented faults can be ascribed to a single strike-slip stress field (R'=1.43), characterized by a subvertical $\sigma_2$ and a $\sigma_1$ oriented 026/08 (Fig. 8A), very similar to the shortening direction required for the formation of the Palazzuolo anticline.

20 Given the current vertical-to-overturned attitude of the beds at the studied outcrop, to exclude the possibility that the refracted faults formed at a different orientation and were only later rotated during the development of the Palazzuolo anticline, we rotated them coherently with the bedding until the latter attained a horizontal attitude, i.e. mimicking a possible pre-anticline setting. The restoration, carried out in progressive steps of 60°, was performed around a horizontal axis striking 130, i.e. parallel to the local average orientation of the Palazzuolo anticline axis (Figs. 8B-8D). Restoration 25 leads to faults compatible with a NW-SE extensional stress field. We deem this unrealistic, however, because non-andersonian faulting conditions are systematically generated. In addition, the derived extensional deformation phase would have ensued during a pre- or syn-folding time and within this portion of the Northern Apennines this is not supported by any independent geological evidence, neither in literature nor from our own field observations. Therefore, we conclude that the studied faults developed within a strike-slip stress field, with $\sigma_1$ oriented 026/08 and no block rotation 30 has taken place since the formation of the structures studied here.

### 4.3 Slip tendency analysis

By using the reduced stress tensor constrained by our paleostress analysis, it was possible to compute slip and dilation values for every orientation and to assess the relative slip and dilation potentials of the structures observed within the study site. As the tendency computations are based on paleostress analysis, it is to be expected that shear fractures attain 35 highest slip tendency values and tensional fractures highest dilation tendency values. The usage of slip and dilation tendency analysis, however, makes it possible to further assess the effect of the stress ratio upon faulting, which, in the extreme case of values close to 0 or 1 may result in counterintuitively high slip and dilation tendency values for specific orientations (e.g. Morris & Ferrill 2009, Leclère & Fabbri 2013). The results of our tendency analysis show that the observed tensile fractures (related to the refracted strike-slip faults, black dots in Fig. 8E) have relatively low slip 40 tendency, but high dilation tendency, with the highest values for fractures with an average attitude of 20/300, which





suggests fracture opening parallel to the interpreted $\sigma_3$ direction (Fig. 8E). As for the sinistral refracted faults (pink dots in Fig. 8E), the steeper ones have an average pole attitude of 315/20 and are characterized by high slip tendency and low dilation tendency (Fig. 8E). Interestingly, Fig. 8E shows areas characterized by relatively large slip and large dilation tendencies, possibly indicating the existence of shear fractures with a relatively large dilation component (white circles in Fig. 8E).

## 5 Discussion

The highly deformed MAF volume to the northeast of the Palazzuolo anticline and the related Mt. Castellaccio thrust contains outstanding examples of refracted FFP that, as revealed by paleostress analysis, developed within a strike-slip tectonic regime with $\sigma_1$ oriented 026/08. While refracted normal FFP are described by numerous studies in the literature,

refracted strike-slip FFP are not described to the best of our knowledge.

Detailed fieldwork coupled with the systematic analysis of the geometric and kinematic characteristics and parameters of the studied FFP and the identification of structures and structural patterns documenting different stages of the local evolution, have allowed us to better understand the mechanical behaviour of the studied strike-slip faults throughout the sedimentary multilayer.

The spatially partitioned coexistence of different fracture modes within the multilayer is best accounted for by a conceptual brittle mechanical model based upon two distinct failure envelopes and related Mohr circles that describe independently (yet contemporaneously) the distinct evolution of the tensile/hybrid and shear fractures within the strong and weak layers, respectively.

Our observations suggest that, following the initial application of a stress field upon the MAF multilayer, failure first

localized in the weak mudstone layers during the build-up of differential stress. This reflects the fact that, while enlarging, the Mohr circle touches first the failure envelope of the mechanically weaker lithotype, i.e. the mudstone (Figs. 5A and 9A). The presence of isolated tension gashes parallel to the $\sigma_1$-$\sigma_2$ plane within some of the exposed sandstone layers cannot be considered as evidence of original nucleation within the strong layers. Our findings suggest them to rather be elements of pre-existing weakness that drive the localization of the tensile component of the refracted faults within strong

layers or even structures belonging to a later deformation stage (in both cases formed after the shear fractures within mudstone layers; Fig. 5A).

Once the shear planes propagate further through the multilayer, the rheological contrast causes the refraction at significant interfaces (i.e., either bed-bed interface or intra-bed surface where there occur significant compositional or grain-size variations) and the local mechanical properties control the overall failure mode. Refraction itself, in fact, consists in a

change of the angle θ between the orientation of the FFP and $\sigma_1$. When θ is close to 0°, failure takes place primarily by tensile fracturing; for θ up to 25°, fracturing becomes hybrid, and for θ > 25° deformation occurs by shear fracturing. The studied refracted faults within strong layers are characterized by a θ angle between 0 and 20°, which allows us to classify them as tensile or hybrid fractures as a function of θ itself. Fault propagation through the sandstone layers occurs by the progressive weakening of the volume of rock at the fault tip through structural processing by dilatant and hybrid fractures

that progressively develop and connect, allowing the growth of the shear fault plane (i.e. a "process zone", *sensu* Fossen, 2010; Fig. 6).

Our detailed mesoscopic observations have led us to conceptualize a model for the mechanical behaviour and evolution of the studied strike-slip refracted faults. We illustrate it in the following by also referring to Fig. 9, where every step of the model describing the progressive evolution of the FFP is illustrated by a Mohr-Coulomb diagram and a picture

representing the corresponding structure/stage as observed in the field:

1-      The studied faults nucleate initially as shear fractures within mudstone layers. Nucleation therein reflects failure during regional and local stress build-up because of the lower mechanical strength of mudstone. The different mechanical



properties of strong and weak layers, at this early stage allow the elastic storage of stress within the sandstone layers before failure, while mudstone layers reach earlier their yield point through a more "plastic" behaviour (Fig. 9A, e.g. Giorgetti et al., 2016).

2-    When the FFP propagates from the weak layer toward adjacent strong layers and reaches the planar layer-layer interface, it refracts because of the different mechanical properties of the stronger sandstone (e.g. Ferrill et al., 2012). The presence of thin calcite films along bed-bed interface-parallel faults suggests a dilational component at high-angle to the layering indicative of a partial decrease of the normal stress, possibly reflecting a decrease of effective stress (i.e. a decrease in mean stress), a condition necessary for the first tensile/hybrid fractures to occur within the strong layers (Fig. 9B). These first fractures testify the progressive formation of a hybrid process zone ahead of the tip of the FFP. Importantly, the presence of calcite infill also in the tensile fractures formed during this incipient stage of deformation of the strong sandstone layers confirms that fluids may indeed have played a role in further lowering the mean stress. Precipitation and crystallization of calcite allows also for a partial recovery of shear stress within the sandstone layers, which leads to more suitable conditions for the formation of hybrid fractures. Unfortunately, we lack clear-cut evidence to establish whether it is tensile or hybrid fractures to develop first. In any case, the well documented coexistence of tensile and hybrid FFP implies that from this stage onwards the deformation of strong layers becomes complex and dynamic, switching cyclically and repeatedly between two different stress conditions. Since shear deformation is still ongoing within the weak layers, a differential mechanical behaviour is required also at the scale of the deforming multilayer, wherein the deformation histories of the two involved lithologies evolve partially independently from each other. This requires adopting two failure envelopes and two related Mohr circles, despite one single regional stress field.

3-    As deformation continues, fragmentation within the process zone increases, creating a complex network of intersecting fractures and dislodging host rock fragments, which end up being trapped within the tensile and hybrid fracture calcite vein infill (Fig. 9C). At this stage, deformation of the process zone is not mature enough to make the fault propagate further to the next weak layer yet.

4-    The process zone becomes then totally disrupted to facilitate enhanced fluid flow, calcite precipitation and to have the FFP cut through the sandstone layer and propagate toward the next weak layer (Fig. 9D). The volume of the dilated process zone infilled by calcite veins becomes significantly larger as strain progressively accumulates.

5-    The final stage is characterized by the development of one major, thoroughgoing fracture within the strong layer, with the dilatant segment in the competent layer fully filled by calcite. The pervasive deformation style of the mature FFP propagation could potentially obliterate all evidence of the earlier deformation stages (fig. 9E).

The shallow crustal level at which deformation occurred (see also Section 2) suggests that the studied refracted strike-slip faults developed under conditions of relatively low differential stress since the beginning of the faulting event. This fact and the lack of structures testifying extreme fluid pressure conditions (e.g. hydraulic breccias) do not allow to estimate to what extent (overpressured?) fluids assisted the deformation process. On the one hand, in fact, (over)pressured fluids at depth migrate vertically along planes tracking the $\sigma_1$ and $\sigma_2$ principal stresses, perpendicularly to $\sigma_3$ within the open fractures, at least partially driven by the negative pressure gradient caused by the opening of the tensile fractures themselves. On the other hand, fluid pressure probably contributes to lower the mean stress enough for tensile and hybrid fractures to form. In both cases, what can be evidenced by our detailed field observations and our conceptual model is that circulation of pressured fluids together with grain size, mineralogy (especially the content of clay minerals) and mechanical properties of the fracturing rock (cohesion, angle of internal friction and tensile strength) fully control the localization, development and propagation of the studied strike-slip, refracted faults.



**6 Conclusions**

Our study of refracted brittle strike-slip FFP within mechanical multilayer sequences has highlighted the major role played by the heterogeneous mechanical properties of alternating strong and weak lithologies in deviating FFP trajectories and causing the localised coexistence of different failure modes. Our study on refracted strike-slip faults aimed to address and

give new constraints on fundamental questions concerning the details of nucleation and propagation of FFP, development of hybrid fractures and the processes governing the transition from one failure mode to the other (tensile, hybrid, and shear). The recognition of clear geological structures describing the incipient and/or intermediate evolution stages of the studied strike-slip refracted FFP has allowed us to conclude that:

-      Nucleation occurs within weak layers;

-      Refraction and its magnitude are mainly controlled by grain-size and content in clay minerals, which in turn steer the mechanical properties such as shear strength and friction coefficient of the involved lithologies;

-      Propagation occurs by the fluid-assisted development of a complex process zone within strong layers in front of the tip of the growing FFP;

-      The process zone is characterized by a network of coexisting tensile and hybrid fractures that evolve, through

progressive weakening and fragmentation of the affected rock volume, into a tensile fracture completely filled by calcite, suggesting a transient cyclical increase and decrease of the local differential stress as failure is accommodated by cyclic tensile and hybrid failure modes;

-      Pressured fluids play an important role in achieving suitable conditions for the development of tensile and hybrid fractures.

The authors declare that they have no conflict of interest.

**Acknowledgments**

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

**Figures captions:**

Fig. 1 – Geological map and cross section of the area crossed by the Santerno stream where the Palazzuolo anticline, the Mt. Castellaccio thrust and the studied area (black rectangle) are located. Modified after Tinterri and Tagliaferri (2015).



Fig. 2 – Outcrop view of the studied MAF layers along the banks of the Santerno stream. The stereonet represents on a Schmidt (lower hemisphere) diagram all the faults families recognised within the study area.

Fig. 3 – Thin section microphotography of the two lithologies (sandstone and mudstone) affected by the studied faults.

Fig. 4 – Orientation of horizontal principal stresses for different stress regimes and how they are identified by the stress ratio (R) and the stress index (R'). For each stress regime, the relation between R' and R is also shown (modified after Delvaux et al., 1997).

Fig. 5 – Field examples of refracted strike-slip faults. Almost all the photos are taken in plan view. Sand. = sandstone, Mud. = mudstone, Cal. = calcite. A) Comprehensive view of a refracted fault with highlighted the presence of vertical tensile gashes and the horsetail geometry of one of the two tips of the fault. B-C) Details about the tensile fractures within the sandstone layers, highlighting the geometry, the refraction angle and how aperture and offset have been measured (represented in fig. 7). D) Example of refracted fault with larger displacement. The coloured areas numbered 1 and 2

indicate different generations of calcite veins, being 1 related to the infill of tensile fractures within sandstone layers and 2 related to the subsequent infill of the more evolved through-going shear fault.

Fig. 6 – Field examples showing evidences of refracted faults at different stages of evolution. Sand. = sandstone, Mud. = mudstone, Cal. = calcite. Tensile fractures are represented in blue, while hybrid fractures are represented in green. A-B)

Well developed shear fault propagating from weak layers (bottom part of the picture) within the strong layers. Here the fractures show a second-order refraction, deformation moves from tensile failure (orthogonal to the bedding) to a branching geometry in correspondence of an abrupt grain-size change (see fig. 5A for location of 6B). C) Another example of evolving refracted faults that propagate through the sandstone layers creating a complex network of calcite-filled fractures and fragmented slices of rock. Colours are added to further help to distinguish sandstone and mudstone layers

and calcite-filled veins.

Fig. 7 – Representation of the relationships between different geometrical parameters measured on the dilational segments of FFP. A) thickness vs aperture of the strong layers; B) thickness vs offset (or displacement) of the sandstone layers; C) aperture vs offset of the dilational fractures. The large scattering is probably due to the fact that the measured parameters

are not normalized to the actual grain-size of the strong layers, see text for explanation.

Fig. 8 – A) Paleostress tensor calculation accounting for the present-day attitude of the left- and right-lateral strike-slip faults, dilatant fractures, reverse faults and main Mt. Castellaccio thrust. The blue and red arrows, together with the principal stress symbols, describe the orientation of the paleostress ellipsoid. The histogram represents the misfit of the

faults attitude with respect to the paleostress tensor. B-D) Rotation of the faults slip data (and related paleostress tensor) with respect to the 130/00-oriented main axis of the Palazzuolo anticline. Each panel represents a rotation of 60°, except the last one (D) that represents a rotation to a complete horizontal attitude of the bedding. E) Results of the slip tendency analysis, the coloured scale represents how prone the measured faults are to reactivation, either in dilation or shear. White circles represent areas of high dilation tendency and high slip tendency, see text for explanation.


Fig. 9 – Multi-stage conceptual mechanical evolution for the formation and propagation of the studied refracted faults, coupled with Mohr diagrams. See text for explanation. Cohesion (C), tensile strength (T) and angle of internal friction (φ) for the sandstone and mudstone lithologies were obtained by laboratory triaxial experiments performed on the MAF





sandstone and mudstone layers (Lembo Fazio et al., 1990). Black failure envelope and Mohr circle refer to sandstone, grey dashed envelope and circle refer to mudstone. The grey areas tentatively represent the differential stress span existing between ideal conditions for development of tensile (lower diff. stress) and hybrid (higher diff. stress) fractures within sandstone layers. During "processing" of the strong sandy layers at the tip of the growing and propagating fracture plane,

5      stress conditions switch cyclically and repeatedly between these two end-members scenarios.



fig. 1



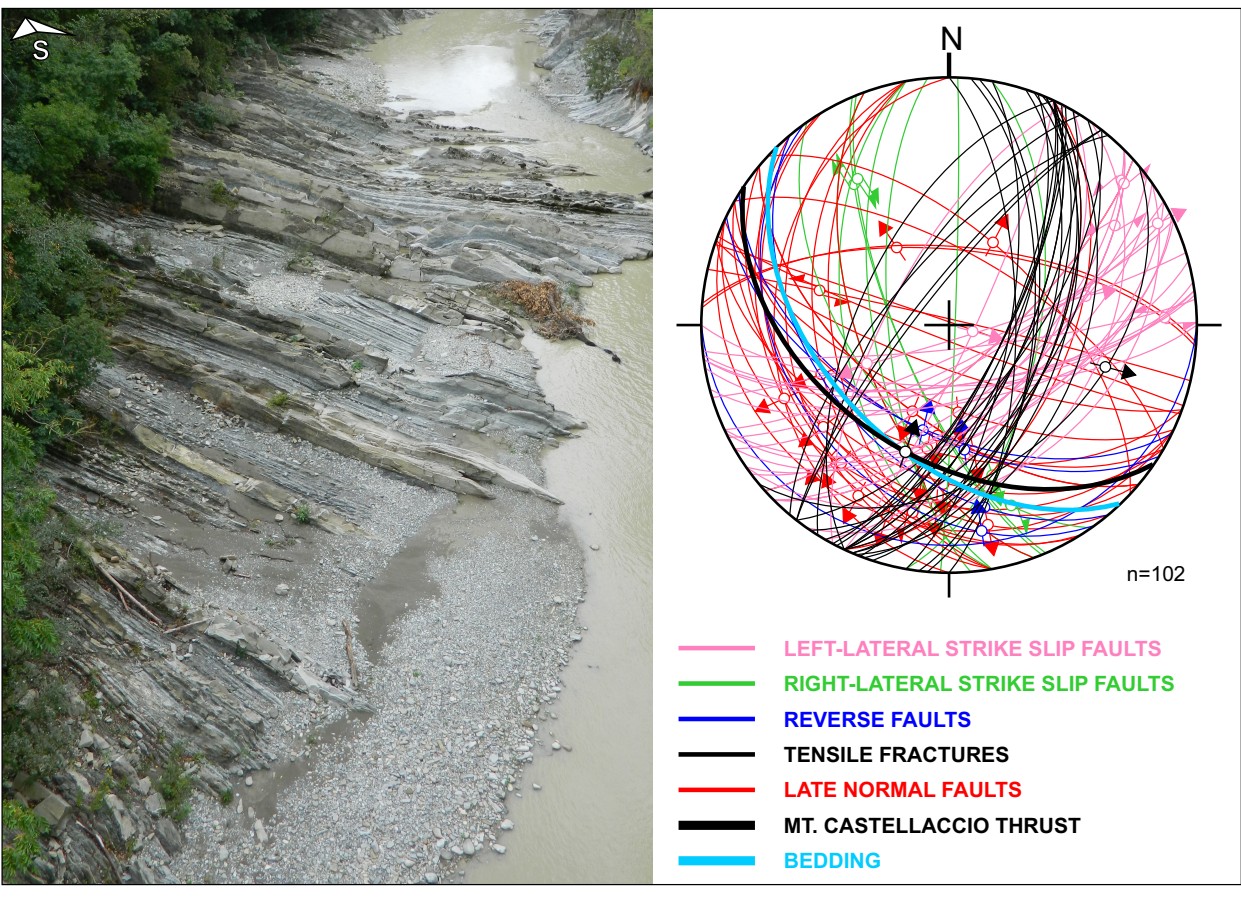

fig. 2



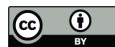

**A - SANDSTONE**

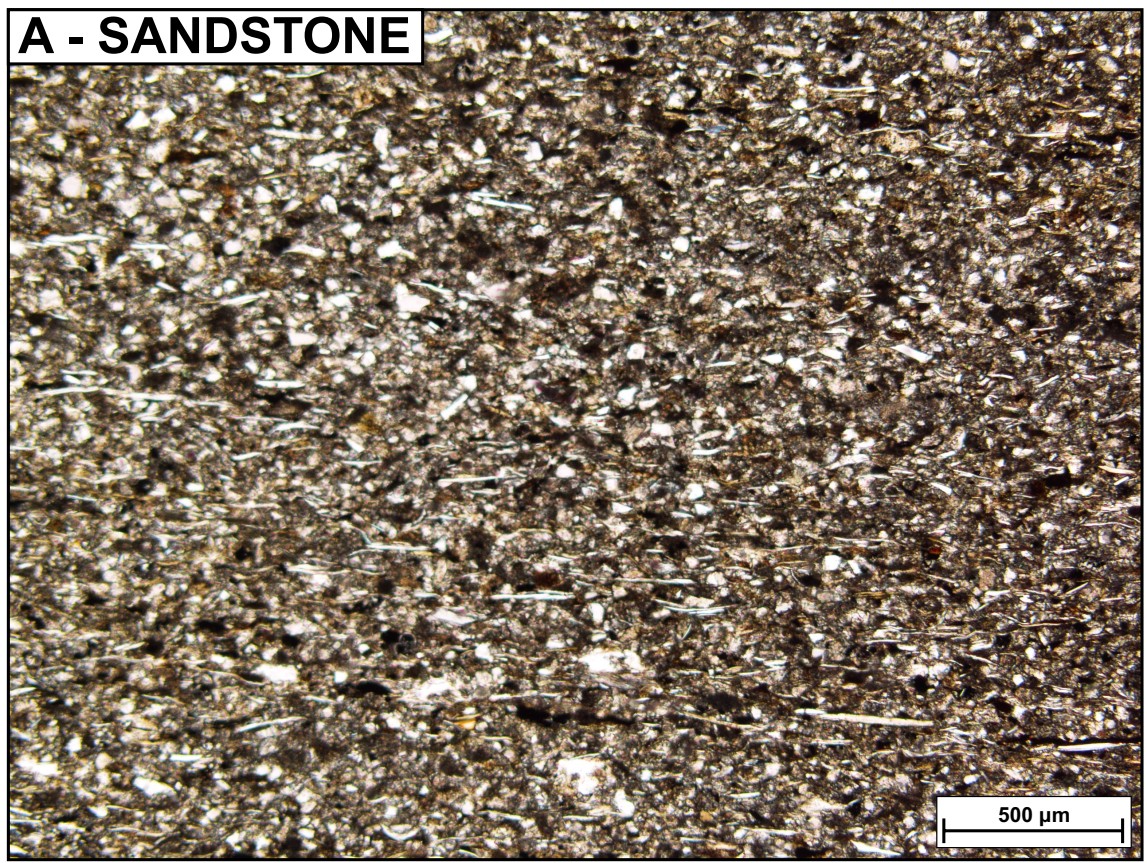

500 μm

**B - MUDSTONE**

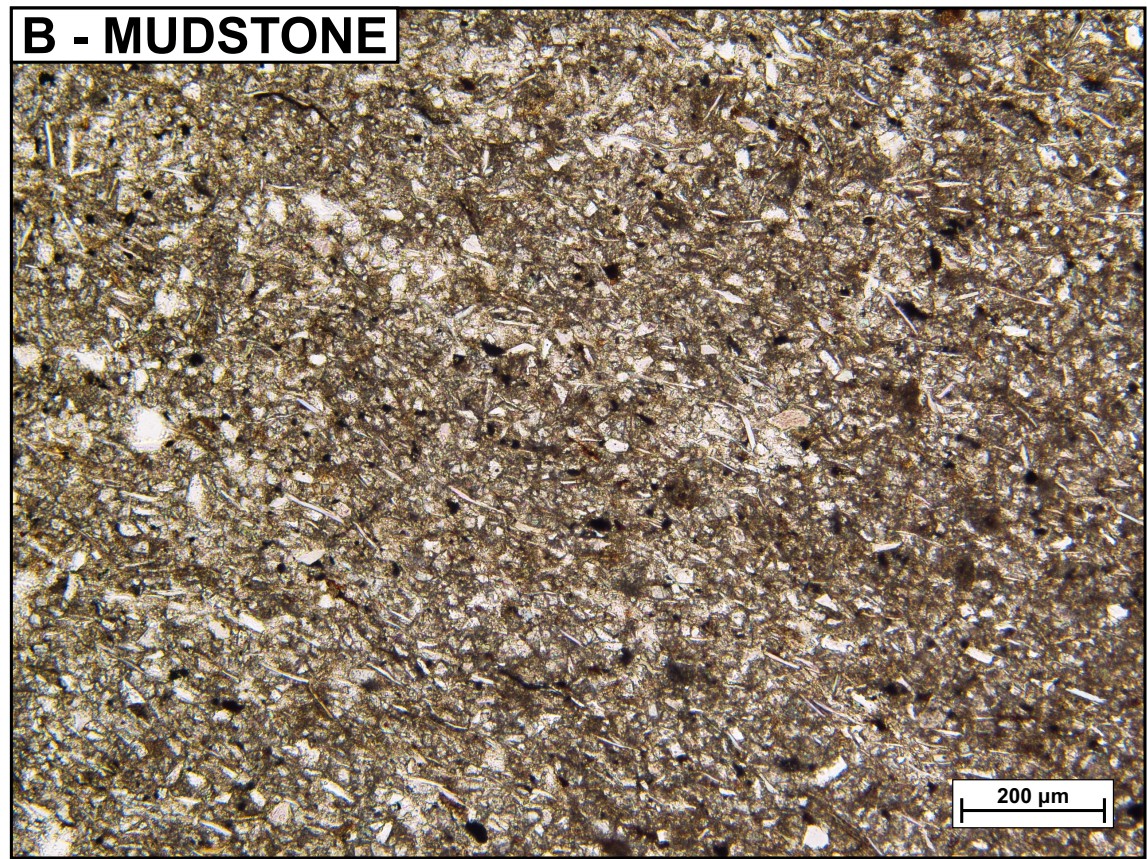

200 μm

fig. 3



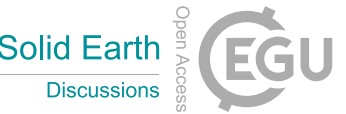

fig. 4

| ORIENTATION OF HORIZONTAL STRESSES | | 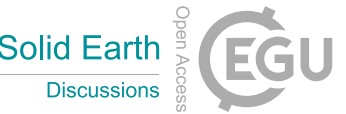 |  |  |  |  |  |  |
|---|---|---|---|---|---|---|---|---|
| STRESS RATIO - R | 0.00 | 0.25 | 0.50 | 0.75 | 1.00 | 0.75 | 0.50 | 0.25 | 0.00 | 0.25 | 0.50 | 0.75 | 1.00 |
| STRESS REGIME | RADIAL EXTENSIONAL | PURE EXTENSIONAL | TRANSTENSIONAL | PURE STRIKE-SLIP | TRANSPRESSIVE | PURE COMPRESSIVE | RADIAL COMPRESSIVE |
| STRESS INDEX - R' | 0.00 | 0.25 | 0.50 | 0.75 | 1.00 | 1.25 | 1.50 | 1.75 | 2.00 | 2.25 | 2.50 | 2.75 | 3.00 |
| DETERMINATION OF R' | R' = R | | | R' = 2-R | | | R' = 2+R | | |





fig. 5



fig. 6





fig. 7





fig. 8



fig. 9