# Peer review of "The role of mechanical stratigraphy on the refraction of strike-slip faults"

_Solid Earth, 2018_

## Referee Comment (RC1) · Anonymous Referee #1 · 22 Aug 2018

In their work Carlini and coworkers present meso-fault data collected in an 200 m wide exposure located in the frontal limb of the Palazzuolo anticline (Apennines, Italy). The observed meso-faults affect overturned strata and are interpreted by the authors as developed, after folding, in a strike-slip stress regime. The meso-faults include segments having different cutoff angles in weak and stiff layers. The authors interpret this cutoff variability as an evidence of refraction of the strike-slip faults during their propagation. They also suggest that strike-slip faults nucleated in weak layers and then propagated across the stiff beds.

The abstract is informative, despite it is too long. The intro is well written and provides an up to date overview of the topic. I have two major concerns about the data and their discussion/interpretation.

[Figure]

1 – The strike slip origin of the faults is, at the best, not convincing. The intersection between the pink left lateral faults and green right lateral faults, as seen in the stereoplot of figure 2, exactly lies along the bedding plane and the striations along the planes of both systems are at 90° from this intersection zone. This could be taken as a textbook example of a pre-folding tilted extensional conjugate system (as shown in figure 8d, where the data are displayed after unfolding). Your paleostress analysis provides slightly different indications (i.e. Sigma 2 not lying along the bedding plane), but this could merely indicate that some of the assumptions behind the stress inversion are wrong (e.g. you have lumped in the inverted dataset structures pertaining to different events). The study of additional outcrops, with different bedding dips, is required to solve the problem: the orientation of the strike slip faults is constant at different bedding dip value = they are post folding; the orientation changes but the angular relationships with the bedding is constant = they are pre to syn folding. As pointed out by the authors, if an extensional origin is assumed the derived extension direction would be NW-SE. i.e. parallel to the trend of the belt. The authors claim that (Page 6 line 27) "the derived extensional deformation phase would have ensued during a pre- or syn-folding time and within this portion of the Northern Apennines this is not supported by any independent geological evidence, neither in literature nor from our own field observations". This is unfair with respect to the thousands of works that have documented pre to syn-folding extension oriented parallel to the trend of foredeep and anticlines in FTBelts (starts with Sterans, 1968; Dietrich, 1989), including the Apennines (Doglioni, 1995).

2 – The idea that faults has nucleated in weak layers is not supported by the presented data. Whatever the origin, post-folding strike-slip or pre-folding extensional, field photographs suggest that fault propagation has occurred throughout the linkage of pre-existing fractures, as commonly observed during fault growth (e.g. Healy et al., 2006). In detail, looking at the photographs, many could say that fault propagation has occurred by the linkage of the pre-existing (or precursory in the case of pre-folding extension) high angle to bedding fractures (those affecting the stiff layers). The authors

suggest a different and counterintuitive history, with the nucleation of faults within the weak layers and propagation (with consequent refraction) in the stiffer layers. Honestly, there is not enough field material to discriminate between the two hypotheses (the presentation of some displacement-distance diagrams could help). At present, the evolutionary model proposed by the authors is based only on an analytical model explained in figure 9: stiff layers require higher tensile stress for failure and, if one assumes that the stress tensor is the same for adjacent layers, failure must occurr before in weak layers and then, after stress build up, in stiff layers. This is a weak argumentation, as the assumption of a constant stress tensor in adjacent layers with different mechanical properties is highly questionable: what about uniaxial or biaxial strain conditions?

---

## Author Comment (AC1) · 21 Sep 2018

We are thankful to the first reviewer for his/her interest in our paper and for his/her help-ful and constructive comment. We will do our best to clarify the parts of the manuscript that, in the light of the reviewer's comments, resulted more confused and/or generated some misunderstanding, we will also enrich the photographic material that helps sup-porting our model. We report below the reviewer's comments followed by the authors' answers.

reviewer: The strike slip origin of the faults is, at the best, not convincing. The in-tersection between the pink left lateral faults and green right lateral faults, as seen in the stereoplot of figure 2, exactly lies along the bedding plane and the striations along

[Figure]

the planes of both systems are at 90 from this intersection zone. This could be taken as a textbook example of a pre-folding tilted extensional conjugate system (as shown in figure 8d, where the data are displayed after unfolding). Your paleostress analysis provides slightly different indications (i.e. Sigma 2 not lying along the bedding plane), but this could merely indicate that some of the assumptions behind the stress inversion are wrong (e.g. you have lumped in the inverted dataset structures pertaining to different events). The study of additional outcrops, with different bedding dips, is required to solve the problem: the orientation of the strike slip faults is constant at different bedding dip value = they are post folding; the orientation changes but the angular relationships with the bedding is constant = they are pre to syn folding. As pointed out by the authors, if an extensional origin is assumed the derived extension direction would be NW-SE. i.e. parallel to the trend of the belt. The authors claim that (Page 6 line 27) "the derived extensional deformation phase would have ensued during a pre- or synfolding time and within this portion of the Northern Apennines this is not supported by any independent geological evidence, neither in literature nor from our own field observations". This is unfair with respect to the thousands of works that have documented pre to syn-folding extension oriented parallel to the trend of foredeep and anticlines in FTBelts (starts with Sterans, 1968; Dietrich, 1989), including the Apennines (Doglioni, 1995)."

authors: The idea that the studied faults cannot be interpreted as rotated normal faults is based on two main arguments: - First, the average dihedral angle between the two conjugated fault families (as identified and suggested by the reviewer) is very large, more than 80°, which would imply strongly non-andersonian conditions during the extensional event; - Second, the kind of faults studied and described in our manuscript are spatially restricted to just an area of few hundreds m2 right at the front of the Palazzuolo anticline, thus strongly suggesting a genetic relationship between their nucleation/development and the shortening that created the fault-related fold. Based on these observations, although we cannot possibly exclude the presence of extensional structures related to pre- to syn-folding extensional tectonics within this portion of the Northern Apennines, we firmly believe that the studied features are indeed strike-slip faults linked to the formation of the antiform, and that they are not indicative of regional-scale, pre- to syn-folding extension or even lithospheric-scale syn-contractional transtension. Concerning this topic, we would like to remark that our speculations were strictly referred to the studied portion of the Northern Apennines, not to the entire chain nor to orogenic wedges in general. In a possible revised version of our manuscript we will add more photographic documentation of the strike-slip kinematics of both the sinistral and dextral faults.

We believe that the reviewer's deduction suggesting that our paleostress analysis may be faulty (derived from his/her idea that the faults should be extensional) is based on a rather circular argumentation, and we invite the reviewer to provide a more direct evidence of the faulty assumptions at the base of the analysis presented by the authors. We have followed a robust scrutiny during the analysis of the fault-slip data and have taken great care while sorting the bulk fault-slip data into mechanically compatible subsets, which were further utilized in the assessment of states of paleostresses.

reviewer: The idea that faults has nucleated in weak layers is not supported by the presented data. Whatever the origin, post-folding strike-slip or pre-folding extensional, field photographs suggest that fault propagation has occurred throughout the linkage of pre-existing fractures, as commonly observed during fault growth (e.g. Healy et al., 2006).In detail, looking at the photographs, many could say that fault propagation has occurred by the linkage of the pre-existing (or precursory in the case of pre-folding extension) high angle to bedding fractures (those affecting the stiff layers).

authors: It is not clear to us what the reviewer has in mind as to which field data should suggest the linkage of pre-existing tensile fractures and not support the nucleation within weak layers. The rare tension gashes observed within strong layers are never observed to link up through by shear fractures within weak layers, while, on the contrary, clear field data show well-developed shear faults within weak layers propagating through strong layers, but without completely cutting through them. We will, yet again, provide better photographic documentation and apologise if our descriptive base in the

paper was note sufficiently clear.

reviewer: The authors suggest a different and counterintuitive history, with the nucleation of faults within the weak layers and propagation (with consequent refraction) in the stiffer layers. Honestly, there is not enough field material to discriminate between the two hypotheses (the presentation of some displacement-distance diagrams could help). At present, the evolutionary model proposed by the authors is based only on an analytical model explained in figure 9: stiff layers require higher tensile stress for failure and, if one assumes that the stress tensor is the same for adjacent layers, failure must occurr before in weak layers and then, after stress build up, in stiff layers. This is a weak argumentation, as the assumption of a constant stress tensor in adjacent layers with different mechanical properties is highly questionable: what about uniaxial or biaxial strain conditions?

authors: We assume a uniform stress applied to the whole multilayer pack, but this initial condition changes immediately, as deformation proceeds. The initial uniform stress, in fact, immediately differentiates, following different paths within the two mechanical layers. This is the core of our proposed conceptual model and we emphasize that we do not assume constant stress tensor except at the very initial state, prior any brittle deformation has taken place. The clear nucleation and propagation structures observed in the studied faults, as mentioned before, do not lend any support to alternative evolutionary conceptual models (such as linkage of pre-existing tensile fractures within stiff layers or even the opposite, that is, linkage of pre-existing shear fractures within weak layers).

---

## Referee Comment (RC2) · Billi (Referee) · 28 Dec 2018

Dear Editor,

I have read with interest the manuscript on the "The role of mechanical stratigraphy on the refraction of strike-slip faults" from the Northern Apennines, Italy. The manuscript is very well written and the science is sound. In my opinion, it could be substantially published as it is. I have, however, a few comments that may contribute to further improve the work by Carlini et alii.

(1) In my opinion, as it is, Fig. 3 is not useful. I suggest Carlini et alii either to omit it or to improve it. In this latter case, more micro-photographs at different enlargements showing various lithological features and details are necessary.

[Figure]

(2) The core of this study is this particular exposure with incipient refracted fractures/faults. I suggest Carlini et alii to further document this exposure and the refracted faults/fractures with some additional photographs.

(4) If I understand well, Carlini et alii exclude that the studied strike-slip faulting are somehow associated with the thrust evolution. I suggest to further discuss/clarify this point.

(3) The study and related results are surely sound and interesting, but I will be frank about my final sensations. After reading it, I do not really understand what I can carry home. In other words, what are the implications (at different scales and in various settings) of these results? And what is the usefulness in tectonics (and other disciplines)? How and where I can use the results provided by Carlini et alii? In synthesis, I suggest Carlini et alii to further discuss the implications and usability of their results in the geosciences?

Sincerely Andrea Billi December 2018

---

## Author Comment (AC2) · 21 Jan 2019

We have much appreciated the review by the second reviewer, Dr. Andrea Billi, who thinks that the paper is clear, well written and scientifically sound. He suggested a few minor changes that we were happy to implement and that we comment on in the following;

The reviewer did not think that the figure with the thin section microphotographs was useful, as it lacked specific details and, perhaps, relevance to the paper.

authors: we have removed the photo. The text describing the mineralogy of the two main lithologies of the Marnoso Arenacea Formation is self-explanatory and we concur that there is no need to document further the lithotypes.

[Figure]

The reviewer asked for the further documentation of the studied faults.

authors: we have done that by introducing Figure 4, as explained above.

The reviewer asked to further clarify the genetic relationships between the faults and the antiform.

authors: we have partially rephrased the text and added some to better explain exactly this point. Having both reviewers commented on this point made us realize that the original version was not clear enough. We are confident that the revised text is now better and explains clearly this crucial point, that is, that the strike-slip faults are cogenetic with the shortening that also caused folding.

The reviewer's last comment pointed to a lack of "relevance" of our study beyond the sheer mechanical aspects of our proposed model.

authors: to show that our results do bear potential implications on a number of structural and tectonic aspects, we have added the following text at the end of our discussion:

"Moreover, our study highlights the details of possible modes of strike-slip fault initiation and development in a sedimentary sequence characterized by layers of contrasting properties and the dynamic stress conditions prevailing during propagation of the faults, which also results in refracted strikes-slip faults. This might be useful to studies dealing with strike-slip faulting at all scales in sedimentary sequences, including the segmentation of strike-slip faults and the formation and development of step-overs as the faults grow by progressively accumulating slip. Fault segmentation (e.g Manighetti et al. 2009) has great implications on the mechanical behaviour of faults and our model provides important constraints on the seismic hazard assessment of such environments. The formation of dilational step-overs also has implications for fluid flow within faults and adjacent rock volumes and our model may provide indications on the location and characteristics of the potential dilational jogs and areas of enhanced per-
meability leading to enhanced fluid flow and mineral precipitation."

Although obviously only a short list of possible applications, the text demonstrates that our study offers indeed interesting inputs and connections to a plethora of structural themes.

---

## Author Response (AR1)

Bologna, January 21, 2019

Dear Editor,

we herewith submit to your attention the revised version of our manuscript "The role of mechanical stratigraphy on the refraction of strike-slip faults" by Carlini and coworkers submitted for possible publication in Solid Earth. We provide you with a file containing all changes highlighted in "review mode" so as to make it easier for you to appreciate what we have modified.

We have followed all critical inputs by the two reviewers who have kindly agreed to review our paper. Back in August we uploaded a first Reply to the comments by reviewer 1. We stand by those replies and in the submitted version that we submit herewith we have made the following changes to improve and clarify the text following that reviewer's input:

- We have prepared a new figure, Figure 4, to expand and improve the photographic documentation of the fault types found and recognised at the outcrop (also requested by reviewer 2).

- We have tried to further tidy up the text and make it shorter when possible.

- We have improved the text so as to better explain the genetic relationships between the strike-slip faults (the main object of our study) and the Palazzuolo Anticline. We are convinced that the strike-slip faults formed as the antiform (which, in turn, is genetically connected to a thrust) nucleated and was progressively amplified. In Section 4.1 (Field data) we have added the following text:

  "*The relative chronology of the faulting events in relation to the development of the Palazzuolo Anticline and the Mt. Castellaccio Thrust remains loosely constrained and difficult to unravel, even though the observed strike-slip faults and reverse faults appear to be syn- to shortly post-anticline. Importantly, we note that the refracted strike-slip faults and the reverse faults are spatially restricted to an area of a few hundreds of m2 just at the front of the Palazzuolo Anticline, thus strongly suggesting a genetic relationship between their nucleation and further development and the shortening responsible for the amplification of the anticline as a fault-related fold.*"

- As to the possibility that the strike-slip faults (sinistral and dextral) actually formed prior to the folding event as conjugate extensional faults (as suggested by Reviewer 1), we have modified the text consistently with the rebuttal letter back in August. The current text reads now like the following:

  "*Given the current vertical-to-overturned attitude of the beds at the studied outcrop, to exclude the possibility that the refracted faults formed at a different orientation and were only later reoriented during the development of the Palazzuolo Anticline, we rotated them coherently with the bedding until the latter attained a horizontal attitude, thus mimicking a possible pre-anticline setting. This restoration, carried out in progressive steps of 60°, was performed around a horizontal axis striking 130, i.e. parallel to the local average orientation of the Palazzuolo Anticline axis (Figs. 8B-8D). In the pre- to syn-Palazzuolo Anticline folding stage (Fig. 8D), the sinistral and dextral strike-slip faults become moderately-dipping top-to the NW and SE extensional faults, respectively, and the tensile fractures strike ENE-WSW. Restoration leads to faults that could thus be the expression of a NW-SE extensional stress field. We note, however, that the average dihedral angle between the two fault families is > 80°, which would imply strongly non-Andersonian conditions during this potential pre-folding extensional event and thus argues against them being conjugate structures. Additionally, the existence of this NW-SE pre- to syn-folding extensional phase is not documented or supported by any*"

*systematic and convincing independent geological evidence for the studied portion of the Northern Apennines. We conclude, therefore, that the refracted strike-slip faults are a local expression of the Palazzuolo Anticline nucleation and amplification and that, as such, are coeval with or post-date the development of the fold. They developed within a strike-slip stress field, with $\sigma_1$ oriented 026/08."*

We think that the new text is clearer and explains well why we exclude the possibility of the passive rotation of original normal faults into their current attitude, associated with an "apparent" strike-slip kinematics. The large dihedral angle is a solid mechanical reason against the possibility that those faults were initially conjugate extensional structures.

We have also better explained the "regional geological" reasons against a pre-folding NW-SE extensional field. We hope that, by rephrasing the text it will now be clear and obvious that we only refer to this portion of the Northern Apennines and its specific structural evolution through time and not to orogenic wedges in general.

We have much appreciated the review by the second reviewer, Dr. Andrea Billi, who thinks that the paper is clear, well written and scientifically sound. He suggested a few minor changes that we were happy to implement and that we comment on in the following;

-   Dr. Billi did not think that the figure with the thin section microphotographs was useful, as it lacked specific details and, perhaps, relevance to the paper.
    We have removed the photo. The text describing the mineralogy of the two main lithologies of the Marnoso Arenacea Formation is self-explanatory and we concur that there is no need to document further the lithotypes.
-   Dr. Billi asked for the further documentation of the studied faults. We have done that by introducing Figure 4, as explained above.
-   Dr. Billi asked to further clarify the genetic relationships between the faults and the antiform. As explained above, we have partially rephrased the text and added some to better explain exactly this point. Having both reviewers commented on this point made us realize that the original version was not clear enough. We are confident that the revised text is now better and explains clearly this crucial point, that is, that the strike-slip faults are cogenetic with the shortening that also caused folding.
-   Dr. Billi's last comment pointed to a lack of "relevance" of our study beyond the sheer mechanical aspects of our proposed model. To show that our results do bear potential implications on a number of structural and tectonic aspects, we have added the following text at the end of our discussion:

    *Moreover, our study highlights the details of possible modes of strike-slip fault initiation and development in a sedimentary sequence characterized by layers of contrasting properties and the dynamic stress conditions prevailing during propagation of the faults, which also results in refracted strikes-slip faults. This might be useful to studies dealing with strike-slip faulting at all scales in sedimentary sequences, including the segmentation of strike-slip faults and the formation and development of step-overs as the faults grow by progressively accumulating slip. Fault segmentation (e.g Manighetti et al. 2009) has great implications on the mechanical behaviour of faults and our model provides important constraints on the seismic hazard assessment of such environments. The formation of dilational step-overs also has implications for fluid flow within faults and adjacent rock volumes and our model may provide indications on the location and characteristics of the potential dilational jogs and areas of enhanced permeability leading to enhanced fluid flow and mineral precipitation.*

    Although obviously only a short list of possible applications, the text demonstrates that our study offers indeed interesting inputs and connections to a plethora of structural themes.

We hope that you will find the revised text suitable publication in Solid Earth. We look forward to hearing back from you at your earlies convenience.

Yours sincerely,

Mirko Carlini

**The role of mechanical stratigraphy on the refraction of strike-slip faults**

Mirko Carlini[1*], Giulio Viola[1], Jussi Mattila[2], Luca Castellucci[1]

[revised manuscript text omitted]

**4 Results**

**4.1 Field data**

Within the analysed area, the MAF deformed in a completely brittle manner and  contains faults that  assign  to five different  families on a geometric and kinematic basis (Figs. 2, 4): (1) SE to SSE steeply-dipping refracted left-lateral strike-slip faults; (2) SW steeply-dipping right-lateral strike-slip faults; (3) SW steeply-to-shallowly dipping reverse faults; (4) SW-dipping bedding-parallel faults and (5) N-to-NE and W-to-SW shallowly-to-steeply dipping low- and high-angle normal faults .

ESE to SE moderately and steeply-dipping dilatant tensile fractures are also present. None of the observed fault families contain a significant damage zone, not even within the weak mudstone layers (Figs. 4, 5). Fault cores within the weak layers are usually very thin, less than 1 cm thick, and are occasionally coated with calcite. For the more mature faults, however, calcite forms a thicker infillsinfill (~1-5 cm; Fig. 5D). In the case of the most significantly refracted faults, the thickness of the fault core changes dramatically as a function of the failure mode observed within the different mechanical layers (see section 4.3 for details; Fig. 5).

The relative chronology of the faulting events in relation to the development of the Palazzuolo Aanticline and the Mt. Castellaccio Tthrust remains loosely constrained and difficult to unravel, even though the observed strike-slip and reverse faults appear to be syn- to shortly post-anticlinealmost all the observed faults appear to postdate these regional structures. Importantly, we note that the refracted strike-slip faults and the reverse faults are spatially restricted to an area of a few hundreds of m$^2$ just at the front of the Palazzuolo Anticline, thus strongly suggesting a genetic relationship between their nucleation and further development and the shortening responsible for the amplification of the anticline as a fault-related fold.

[revised manuscript text omitted]
). In the pre- to syn Palazzuolo Anticline folding stage (Fig. 8D), the sinistral and dextral strike-slip faults become moderately-dipping top-to-the NW and SE extensional faults, respectively, and the tensile fractures strike ENE-WSW. Restoration leads to faults that could thus be the expression of a NW-SE extensional stress field. We note, however, that the average dihedral angle between the two fault families is > 80°, which would imply strongly non-Andersonian conditions during this potential pre-folding extensional event and thus argues against them being conjugate structures.

are systematically generated. In additionAdditionally, the the existence of this NW-SE pre- to syn-folding derived extensional deformation phase would have ensued during a pre- or syn-folding time and within this portion of the Northern Apennines this is not documented or supported by any systematic and convincing independent geological evidence for the studied portion of the Northern Apennines, neither in literature nor from our own field observations. Therefore, weWe conclude, therefore, that the studied faults the refracted strike-slip faults are a local expression of the Palazzuolo Anticline nucleation and amplification and that, as such, are coeval with or post-date the development of the fold. They 
[revised manuscript text omitted]

Even though the final geometry and some of the mechanical aspects steering the evolution of the studied strike-slip faults are  similar to those described in the literature for normal faults, a straightforward comparison of our conceptual model to the classic schemes for normal faults summarized in Section 1 is not possible because:

- the strike-slip tectonic regime implies important differences in the orientation of both stress and strain ellipsoids;
-  intermediate stages of development of refracted faults are rarely observed in generally high-displacement extensional faults, leaving uncertainties about the details of initial strain localization and deformation history of refracting faults and fractures.

In summary, our results help therefore to clarify important details of the intermediate steps of the evolution of refracted faults and fractures, indicating an alternative mechanical model to that generally proposed for extensional faults. Moreover, our study highlights the details of possible modes of strike-slip fault initiation and development in a sedimentary sequence characterized by layers of contrasting properties and the dynamic stress conditions prevailing during propagation of the faults, which also results in refracted strikes-slip faults. This might be useful to studies dealing with strike-slip faulting at all scales in sedimentary sequences, including the segmentation of strike-slip faults and the formation and development of step-overs as the faults grow by progressively accumulating slip. Fault segmentation (e.g Manighetti et al. 2009) has great implications on the mechanical behaviour of faults and our model provides important constraints on the seismic hazard assessment of such environments. The formation of dilational step-overs also has implications for fluid flow within faults and adjacent rock volumes and our model may provide indications on the location and characteristics of the potential dilational jogs and areas of enhanced permeability leading to enhanced fluid flow and mineral precipitation.

**6 Conclusions**

Our study on refracted strike-slip faults aimed to address and give new constraints on fundamental questions concerning the details of nucleation and propagation of FFP, development of hybrid fractures and the processes governing the transition between different fracturing modes (tensile, hybrid and shear).  It has highlighted the  crucial role played by the heterogeneous mechanical properties of alternating strong and weak lithologies in deviating FFP trajectories and causing the localised coexistence of different failure modes.  The recognition of clear geological structures describing the incipient and/or intermediate evolution stages of the studied strike-slip refracted FFP has allowed us to conclude that:

- Nucleation occurs within weak layers;

- Refraction and its magnitude are mainly controlled by grain-size and content in clay minerals, which in turn steer  mechanical properties such as shear and tensile strength and friction coefficient of the involved lithologies;

[revised manuscript text omitted]

5   Tinterri, R. and Tagliaferri, A.: The syntectonic evolution of foredeep turbidites related to basin segmentation: Facies response to the increase in tectonic confinement (Marnoso-arenacea Formation, Miocene, Northern Apennines, Italy), Mar. Pet. Geol., 67, 81–110, doi:10.1016/j.marpetgeo.2015.04.006, 2015.

**Figures captions:**

Fig. 1 – Geological map and cross section of the area crossed by the Santerno stream where the Palazzuolo anticline, the Mt. Castellaccio thrust and the studied area (black rectangle) are located. Modified after Tinterri and Tagliaferri (2015).

Fig. 2 – Outcrop view of the studied MAF layers along the banks of the Santerno stream. The stereonet represents on a Schmidt (lower hemisphere) diagram all the five main faults families recognised within the study area. In addition, tensile fractures are plotted separated from the fault family they belong to (left-lateral strike-slip faults), in support to the topics discussed hereafter.

Fig. 3 – Orientation of horizontal principal stresses for different stress regimes and how they are identified by the stress ratio (R) and the stress index (R'). For each stress regime, the relation between R' and R is also shown (modified after Delvaux et al., 1997). Thin section microphotography of the two lithologies (sandstone and mudstone) affected by the studied faults.

20

Fig. 4 – Field examples of the main fault families recognised within the study area. A) Steeply-dipping refracted left-lateral strike-slip faults. B) Steeply-dipping right-lateral strike-slip faults. C) Steeply-to-shallowly dipping reverse faults. D) Bedding-parallel faults crosscutting and generally post-dating strike-slip faults. E) Shallowly-to-steeply dipping low- and high-angle normal faults. Orientation of horizontal principal stresses for different stress regimes and how they are

[revised manuscript text omitted]